# Venous Thrombosis Due to Duplication of the Inferior Vena Cava with Decreased Protein C Activity: A Case Report

**DOI:** 10.3390/medicina59030605

**Published:** 2023-03-18

**Authors:** Yukiya Okada, Tsuneaki Kenzaka

**Affiliations:** 1Department of General Medicine, Muraoka Public Hospital, Kami-cho 667-1311, Japan; 2Division of Community Medicine and Career Development, Kobe University Graduate School of Medicine, 2-1-5, Arata-cho, Hyogo-ku, Kobe 652-0032, Japan

**Keywords:** duplication of the inferior vena cava, venous malformation, deep vein thrombosis, protein C, coagulation factors, anticoagulation therapy

## Abstract

*Introduction*: Duplication of the inferior vena cava (IVC) is a congenital venous malformation that occurs in 0.2%–3% of the population as a result of persistent left and right supracardinal veins. The IVC duplication is prone to deep vein thrombosis due to endothelial dysfunction and associated venous stasis. This is a rare case of recurrent venous thrombosis due to IVC duplication and decreased protein C activity. *Case*: A 57-year-old male presented with swelling of the left lower limb that gradually developed over a one-week period preceding his visit. He reported a history of superior mesenteric vein thrombosis, approximately three years ago, for which he received anticoagulation therapy for three months. Thoracoabdominal contrast-enhanced computed tomography (CT) revealed thrombi in the locations of the bilateral main pulmonary arteries, IVC duplication, left common iliac vein, left IVC, and left renal vein. Blood work confirmed protein C activity of 21% (baseline 64%–146%), that could have contributed to the recurrent IVC thrombosis and formation of pulmonary artery thrombus. Subsequently, the patient was hospitalized and started on anticoagulation therapy. The swelling in the left lower extremity gradually improved, and the patient was instructed to continue anticoagulation therapy permanently. *Conclusion*: When investigating venous thrombosis of unknown or recurrent origin, it is necessary to include venous malformations and abnormal activity of blood coagulation factors in differential diagnosis.

## 1. Introduction

Duplication of the inferior vena cava (IVC) is a congenital venous malformation that occurs in 0.2%–3% of the population and forms as a result of persistent left and right supracardinal veins [1]. It is usually asymptomatic and presents as an incidental finding on imaging studies [2]. It has been reported that IVC malformations are associated with 5% of deep vein thrombosis cases [3] and that IVC duplication is prone to deep vein thrombosis owing to venous stasis and associated endothelial dysfunction [3].

The main risk factors for deep vein thrombosis are stasis, hypercoagulability, and vessel wall injury, known as Virchow’s triad [4].

Stasis and vessel wall injury are more likely to occur on the ground of IVC duplication [3]. Hypercoagulability could result from a decrease in protein S or C activity. In rare cases, venous thrombosis can occur due to the combined effects of IVC duplication and decreased protein S [5] or protein C [6] activity.

In this study, we report a case of venous thrombosis due to IVC duplication with decreased protein C activity.

## 2. Detailed Case Description

A 57-year-old Japanese man presented to our hospital with a chief complaint of swelling in the lower left extremity. His medical history included chronic kidney disease, hypertension, hyperuricemia, and dyslipidemia. He had developed superior mesenteric vein thrombosis approximately three years ago and had received anticoagulation therapy with heparin and apixaban for three months. Subsequent anticoagulation therapy was not suggested by his physician at the time. Following the discontinuation of treatment, the patient did not report any symptoms of thrombotic phenomena until the week before his visit to our hospital. During his visit, he reported that the swelling of the left lower limb had been gradually getting worse over the past week, which compelled him to seek medical attention. The patient did not have any known family history of thrombosis, was a smoker (37 pack-years), had no anticoagulants in his medication regimen at the time of admission, and had a sedentary lifestyle—he was a long-distance truck driver.

### 2.1. Investigations

During physical examination, the state of consciousness was normal, blood pressure was 142/90 mmHg, pulse was 98 beats/min, temperature was 36.8 °C, respiratory rate was 22 breaths/min, and SpO_2_ was 97% (room air). The jugular vein was not distended, heart sounds were normal with no audible murmurs, and respiratory sounds were clear. The abdomen was flat and soft with no tenderness, and a surgical scar was observed around the umbilicus. The patient had swollen and slightly cyanotic left lower extremity. The dorsal pedal arteries were palpable bilaterally.

A blood sample was collected during admission, and the bloodwork results revealed the following: platelets 11.5 × 10^4^/µL, BUN 29.3 mg/dL, creatinine 2.18 mg/dL, CRP 1.06 mg/dL, D-dimer 31.2 μg/mL, and protein C activity 21% (reference value 64%–146%) (Table 1). No further laboratory workup for factors of hypercoagulability, such as Factor V Leiden, homocystinemia, prothrombin variant mutation of G20210A, and increased Factor Ⅷ levels was performed.

On the 12-lead ECG, heart rate of 98 beats/min, sinus rhythm, a positive axis, clear Q waves, and negative T waves during induction III were observed. Chest radiography (Figure 1) demonstrated slightly increased permeability in the bilateral middle lung fields. In transthoracic echocardiography (Figure 2), both left and right ventricles were of normal in size, as seen on the parasternal left border long-axis tomogram and parasternal left border short-axis tomogram, with no evidence of left ventricular compression. In the four-chamber view, the right ventricle seemed to be slightly enlarged. The tricuspid valve systolic pressure gradient was 22.5 mmHg. Chest contrast-enhanced CT revealed thrombi in the bilateral main pulmonary arteries (Figure 3). Abdominal contrast-enhanced CT revealed thrombi in the IVC duplication, left common iliac vein, left IVC, and left renal vein (Figure 4). The left IVC was slightly smaller in diameter than the normal IVC (right IVC). It branched from the left renal vein and descended alongside the left kidney to contact the left common iliac vein. A tributary branch to the right IVC was observed immediately before it joined the left common iliac vein. The left femoral vein similarly demonstrated a few low-absorption areas, suggestive of thrombi.

### 2.2. Differential Diagnosis and Treatment

The patient was diagnosed with deep vein and pulmonary artery thrombosis associated with IVC duplication. He was hospitalized and started on treatment with oral rivaroxaban at 30 mg per day and was eventually discharged seven days later, after the D-dimer levels, an indicator of thrombotic activity, had decreased. Following the initial period of hospitalization, treatment with oral rivaroxaban at 30 mg per day was continued, and the swelling in the left lower extremity gradually improved. Percutaneous thrombectomy was considered in case anticoagulation therapy did not succeed in resolving the venous thrombus; however, treatment with anticoagulation therapy was eventually successful and percutaneous thrombectomy was not necessary.

### 2.3. Outcome and Follow-Up

Protein C activity, measured 18 days after discharge, was 28% (baseline 64%–146%) and the protein C antigen level was 71% (baseline 70%–150%)—at the lower end of the reference range. As of this day, approximately one year later, the patient is still being treated with rivaroxaban in an outpatient setting, and no recurrent thrombosis has been observed.

## 3. Discussion

We presented a case of recurrent venous thrombosis due to IVC duplication on the grounds of decreased protein C activity. To the best of our knowledge, only one such case has ever been reported [6]. Thus, this constitutes an extremely rare case [4].

The IVC is a large retroperitoneal vein located primarily in the abdomen. It is formed by the confluence of the right and left common iliac veins, which happens approximately at the level of the 5th lumbar. Subsequently, the formed vessel passes through the vena cava foramen magnum of the diaphragm, enters the thoracic cavity and joins the right atrium. Duplication of the IVC occurs owing to the failure of the left supracardinal vein to regress, resulting in the caudal anastomosis of the left supracardinal vein during fetal life [7]. In the current case, the thrombus formed in the left side IVC, i.e., the excessive side, but there have been reports on thrombus formation in the right side IVC [8], suggesting that venous thrombosis can occur on either the normal or excessive IVC side in the duplication of the IVC. Additionally, in this patient, the left IVC was less developed than the right IVC; however, this is not always the case. In other words, the difference in development of the left and right IVC varies across different cases, and a thrombus can form on either side of the IVC.

Moreover, we revisited the patient’s thoracoabdominal contrast-enhanced CT scan that was performed at the time the patient had developed superior mesenteric vein thrombosis and confirmed that the IVC duplication could have been observed even three years ago.

However, at that time, the medical staff in charge did not recognized the IVC duplication. Morita et al.’s classification of the pelvic venous variation of IVC anomalies [1] classifies them into eight types: Type 1, Type 2a–2e, Type 3, and Type 4. Type 2 describes the duplication of the IVC. In the case presented, a traffic branch originating from the left common iliac vein and connecting to the right IVC, corresponding to Type 2b, could be detected [1]. It has been reported that malformations of the IVC are associated with 5% of deep vein thrombosis cases [3]. Deep vein thrombosis associated specifically with IVC duplication has also been reported, but not as frequently in the English literature [9]. The most frequent IVC malformation in Morita et al.’s classification is Type 2a [1]. However, the frequency of venous thrombosis complications of each type is unknown. Deep vein thrombosis associated with IVC duplication is caused by stasis and vessel wall injury [3]. In this case, the angles of the vessels bridging the right IVC to the left IVC, the left common iliac vein to the left IVC, and the left IVC to the left renal vein were sharp, which may have caused stasis of blood flow. In the duplication of the IVC, the increase in venous pressure due to venous stasis and the direct injury by the resulting thrombus can cause endothelial damage. Additionally, in the case presented, protein C activity was measured three times; on the day of admission, 18 days after discharge, and approximately three years ago when the patient had developed superior mesenteric vein thrombosis. The activity levels were all in the low 20% range with or without anticoagulation therapy. These findings lead us to believe that both the decreased protein C activity and IVC duplication contributed to the formation of recurrent venous thrombi.

Therefore, we concluded that the patient needs to be on lifelong anticoagulation therapy. However, a duplication of the IVC found incidentally on imaging studies without a history of venous thrombosis does not necessitate anticoagulation therapy and should be followed up. There are no clear reports on whether anticoagulants should be administered to prevent venous thrombosis in all cases of IVC duplication. Moreover, of note is that protein C activity was measured at 28% (reference value 64%–146%); however, protein C antigen levels were measured at the lower end of the reference range at 71% (reference value 70%–150%). Consequently, the patient was considered to have a congenital protein C abnormality. Because the use of warfarin increases the risk of developing warfarin-induced skin necrosis in patients with decreased protein C activity, only direct oral anticoagulants that do not suppress protein C production were used [10].

The indications for IVC filters in this patient are discussed below. IVC filters are implanted to prevent pulmonary thromboembolism [11]; however, the patient had already developed pulmonary thromboembolism. Moreover, the IVC filter placement was avoided in this patient because of the risk of the endothelial/ vascular injury caused by the thrombus could be potentiated by the placement of the IVC filtered. It should be highlighted that the first-line choice for the prevention and treatment of pulmonary thromboembolism and venous thrombosis is anticoagulation; the IVC filter is a complementary medical device [12].

In this patient, contrast-enhanced CT revealed the duplication of the IVC, but we believe that simultaneously checking the IVC with four-point ultrasonography can achieve non-invasive detection of the duplication of the IVC. Examination with even the slightest assumption of the presence of venous malformations, such as duplication of the IVC, increases the likelihood of detecting findings that may be the cause of venous thrombosis.

## 4. Conclusions

We presented an extremely rare case of recurrent venous thrombosis owing to IVC duplication and decreased levels of protein C activity. When investigating venous thrombosis of unknown or recurrent origin, it is necessary to include venous malformations, such as duplication of the IVC, and abnormal activity of blood coagulation factors, such as decreased protein S and C activity, in the differential diagnosis.

## Figures and Tables

**Figure 1 medicina-59-00605-f001:**
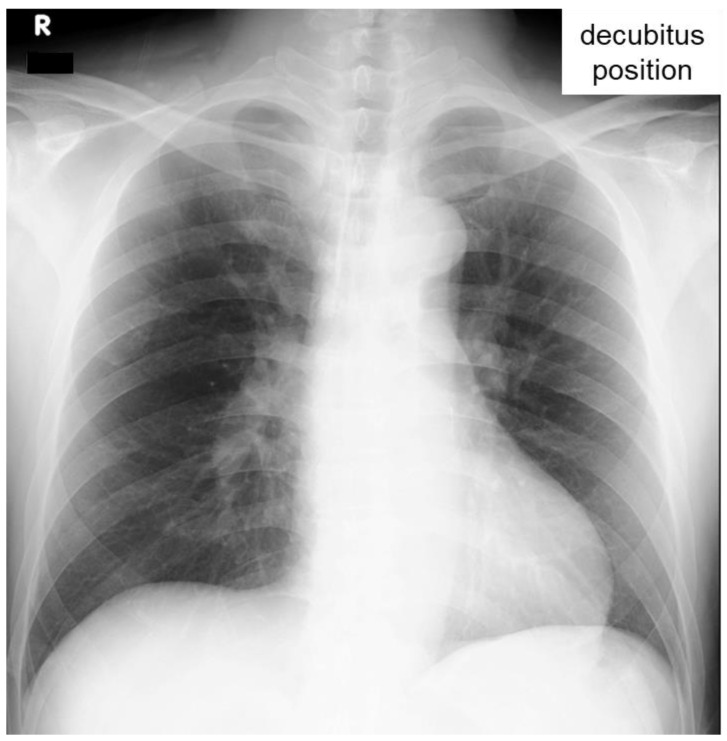
Chest radiograph. Patient’s chest radiography performed on admission and demonstrating slightly increased permeability in the middle lung fields bilaterally.

**Figure 2 medicina-59-00605-f002:**
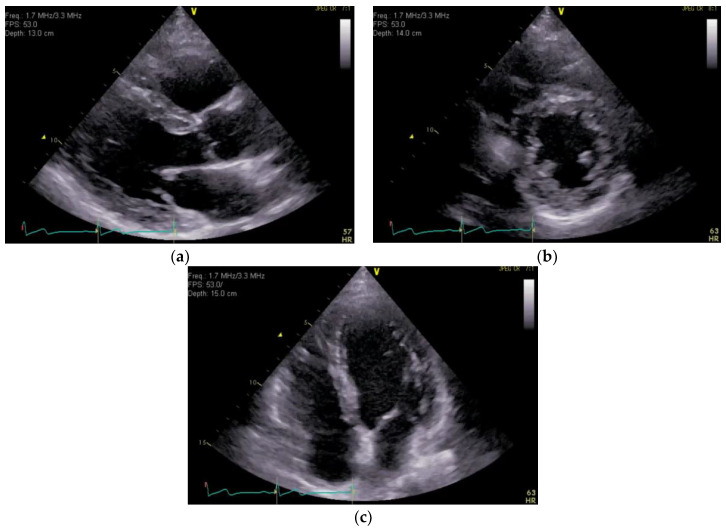
Transthoracic echocardiography. Transthoracic echocardiography showed that both left and right ventricles were of normal size, as seen on the parasternal left border long-axis tomogram (**a**) and parasternal left border short-axis tomogram (**b**), with no evidence of left ventricular compression. In the four-chamber view (**c**), the right ventricle seemed to be slightly enlarged.

**Figure 3 medicina-59-00605-f003:**
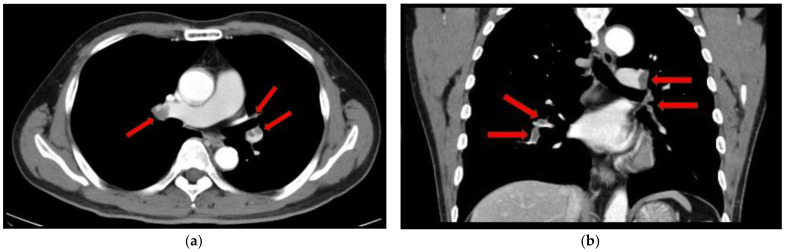
Chest contrast-enhanced computed tomography scan. Thrombi in the main trunk of the bilateral pulmonary arteries. (**a**) Axial view. (**b**) Coronal view. 
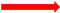
 thrombus.

**Figure 4 medicina-59-00605-f004:**
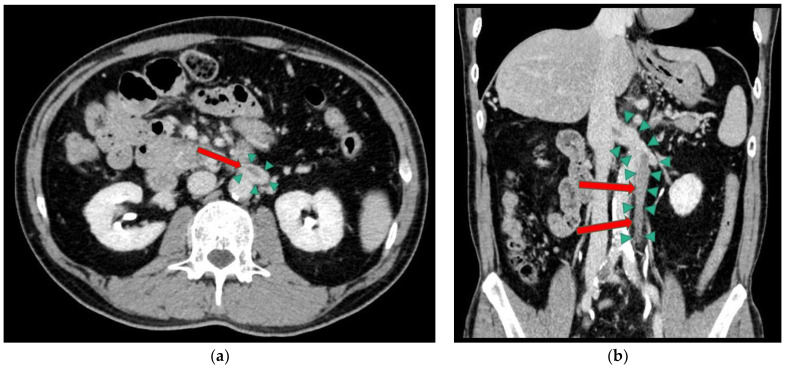
Abdominal contrast-enhanced computed tomography scan. Thrombi in the duplication of the inferior vena cava (IVC), left common iliac vein, left IVC, and left renal vein. (**a**) Axial view. (**b**) Coronal view. 
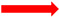
 thrombus. 
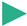
 duplication of the IVC.

**Table 1 medicina-59-00605-t001:** Laboratory data upon admission.

Parameter	Recorded Value	Standard Value
White blood cell count	8000/µL	4500–7500/µL
Neutrophils	74.1%	42%–74%
Lymphocytes	17.6%	18%–50%
Monocytes	7.1%	1%–10%
Hemoglobin	13.2 g/dL	11.3–15.2 g/dL
Platelet count	11.5 × 10^4^/µL	13–35 × 10^4^/µL
Prothrombin time/International normalized ratio	1.02	0.80–1.20
Activated partial thromboplastin time	25.6 s	26.9–38.1 s
D-dimer	31.2 μg/mL	<1.0 μg/mL
Antithrombin Ⅲ	84%	80%–130%
Protein C activity	21%	64%–146%
Protein S activity	119%	67%–164%
C-reactive protein	1.06 mg/L	≤0.60 mg/dL
Total protein	7.5 g/dL	6.9–8.4 g/dL
Albumin	4.2 g/dL	3.9–5.1 g/dL
Total bilirubin	1.2 mg/dL	0.2–1.2 mg/dL
Aspartate aminotransferase	16 U/L	11–30 U/L
Alanine aminotransferase	11 U/L	4–30 U/L
Lactase dehydrogenase	199 U/L	109–216 U/L
Creatine kinase	219 U/L	43–272 U/L
Blood urea nitrogen	29.3 mg/dL	8–20 mg/dL
Creatinine	2.18 mg/dL	0.63–1.03 mg/dL
Sodium	141 mEq/L	136–148 mEq/L
Potassium	4.3 mEq/L	3.6–5.0 mEq/L
Chloride	106 mEq/L	98–108 mEq/L
Glucose	102 mg/dL	70–109 mg/dL
Hemoglobin A1c	5.9%	5.6%–5.9%
Troponin I	5 pg/mL	≤26.2 pg/mL
Brain natriuretic peptide	5.8 pg/mL	≤18.4 pg/mL

## Data Availability

Data sharing is not applicable to this article, as no datasets were generated or analyzed during the current study.

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
