# Peer review of "Venous Thrombosis Due to Duplication of the Inferior Vena Cava with Decreased Protein C Activity: A Case Report"

_medicina, 2023, doi:10.3390/medicina59030605_

Round 1

Reviewer 1 Report

With pleasure, I read the paper titled: “Venous thrombosis due to duplication of the inferior vena cava with decreased protein C activity: A case report”. Overall, the manuscript reads very well. The pertinent history taking, physical examination, investigations, management, and prognosis details were reported. The figures depict the core of the case report. The references are appropriate and language is decent. However, I have the following comments:

1.      It is my understanding that the high risk of deep venous thrombosis is largely due blood flow stasis. Not sure if endothelial dysfunction (as claimed in this case report) is true. Please double-check this piece of information.

2.      Were other causes of hypercoagulability investigated, such as factor V Leiden, homocystinemia, prothrombin variant mutation of G20210A, and increased factor VIII levels?

3.      In Figure 4, please clearly indicate the location of the duplicated IVC with a color different than that of the thrombus. Also, please provide more information about the diameter of both IVC vessels (equal or one is higher/smaller than the other); what is the type of duplicated IVC: prerenal, renal, postrenal, or multisegmental; was there any connection between the iliac vein and the contralateral IVC?

4.      Please elaborate on why IVC filter insertion was not indicated in this case report.  

5.      Patients with duplicated IVC and no evidence of hypercoagulability, can they still develop deep venous thrombosis? Or, an underlying hypercoagulable risk factor must be present along with duplicated IVC to develop deep venous thrombosis?

6.      The discussion section should be enriched with large-sized series literature about the topic. Examples include: PMID: 26221367, PMID: 27666807, PMID: 36495583, and PMID: 28145872

7.      A brief embryological perspective about the potential cause of the anomaly will be helpful to the reader/

8.      Please enrich the discussion section with a brief address on the diagnostic modalities that can help identifying the duplicated IVC besides CT contrast-enhanced imaging.

Author Response

Reviewer 1:

With pleasure, I read the paper titled: “Venous thrombosis due to duplication of the inferior vena cava with decreased protein C activity: A case report”. Overall, the manuscript reads very well. The pertinent history taking, physical examination, investigations, management, and prognosis details were reported. The figures depict the core of the case report. The references are appropriate and language is decent. However, I have the following comments:

 Response:

We would like to express our gratitude to the reviewer. We truly appreciate the constructive and insightful suggestions that have helped us improve our manuscript. We have made every effort to address the issues raised and respond to all comments. The revisions are indicated in red font in the revised manuscript, and we hope that they meet the reviewer’s expectations.

  1. It is my understanding that the high risk of deep venous thrombosis is largely due blood flow stasis. Not sure if endothelial dysfunction (as claimed in this case report) is true. Please double-check this piece of information.

We thank the Reviewer for their constructive comment. In the duplication of the IVC, the increase in venous pressure due to venous stasis and the direct injury by the resulting thrombus can cause endothelial damage. (Lines 177-179)

  1. Were other causes of hypercoagulability investigated, such as factor V Leiden, homocystinemia, prothrombin variant mutation of G20210A, and increased factor VIII levels?

We thank the Reviewer for their insightful question. No further laboratory workup for factors causing hypercoagulability, such as factor V Leiden, homocystinemia, prothrombin variant mutation of G20210A, and increased factor VIII levels was performed. Please note that we made the corresponding addition in-text.(Lines 74-77)

  1. In Figure 4, please clearly indicate the location of the duplicated IVC with a color different than that of the thrombus. Also, please provide more information about the diameter of both IVC vessels (equal or one is higher/smaller than the other); what is the type of duplicated IVC: prerenal, renal, postrenal, or multisegmental; was there any connection between the iliac vein and the contralateral IVC?

We thank the Reviewer for their constructive comment. As per your recommendations, Figure 4 has been revised, and additional information about the duplicated IVC has been included in the text. (Lines 90-93)

  1. Please elaborate on why IVC filter insertion was not indicated in this case report.

We thank the Reviewer for their constructive suggestion. IVC filters are implanted to prevent pulmonary thromboembolism [10]; however, the patient had already developed pulmonary thromboembolism. Moreover, the IVC filter placement was avoided in this patient due to the risk of endothelial injury from the thrombus filling the duplication of the IVC that could be potentiated by the IVC filter. It should be highlighted that the first line choice for the prevention and treatment of pulmonary thromboembolism and venous thrombosis is anticoagulation; the IVC filter is a complementary medical device. The corresponding answer can be found in-text.(Lines 197-201)

  1. Patients with duplicated IVC and no evidence of hypercoagulability, can they still develop deep venous thrombosis? Or, an underlying hypercoagulable risk factor must be present along with duplicated IVC to develop deep venous thrombosis?

We thank the Reviewer for their insightful question. We believe that the duplication of the IVC alone can be a risk factor for thrombosis, and that the concomitant coagulation abnormalities can further promote thrombus formation.

  1. The discussion section should be enriched with large-sized series literature about the topic. Examples include: PMID: 26221367, PMID: 27666807, PMID: 36495583, and PMID: 28145872

We thank the Reviewer for their constructive suggestion and for providing the literature as well. In line with your suggestion, we have included relevant information in the revise manuscript. (Lines 155-161)

  1. A brief embryological perspective about the potential cause of the anomaly will be helpful to the reader/

We thank the Reviewer for their constructive suggestion. Accordingly, we have made in-text additions providing a developmental perspective. (Lines 150-155)

  1. Please enrich the discussion section with a brief address on the diagnostic modalities that can help identifying the duplicated IVC besides CT contrast-enhanced imaging.

We thank the Reviewer for their constructive suggestion. We believe that simultaneously checking the IVC with four-point ultrasonography can achieve non-invasive detection of the duplication of the IVC. We have made corresponding in-text revisions per your suggestion.(Lines 205-209)

Reviewer 2 Report

The authors present a unique case of combination of duplication of IVC and protein C deficiency

Some questions remain unanswered

1. Please detail evidence/ data for the use of rivaroxaban in patients with protein C deficiency - in general coumadin has been used

2. Why was the dosage of rivaroxaban 30 mg and not 20 mg as indicated in the drug package insert?

3. Was percutaneous embolectomy considered considering extensive thrombus burden?

4. Please check for minor grammatical errors and ensure the tense (present/past) is uniform throughout

5. The arrows in the figures seem to be out of order - please ensure they are reflected appropriately

Author Response

The authors present a unique case of combination of duplication of IVC and protein C deficiency

Some questions remain unanswered

 Response:

We would like to express our gratitude to the Reviewer. We truly appreciate the constructive and insightful suggestions that have helped us improve our manuscript. We have made every effort to address the issues raised and respond to all comments. The revisions are indicated in red font in the revised manuscript and we hope that they meet the Reviewer’s expectations.

  1. Please detail evidence/ data for the use of rivaroxaban in patients with protein C deficiency - in general coumadin has been used

We thank the Reviewer for their constructive suggestion. It has been previously reported that the use of warfarin increases the risk of developing warfarin-induced skin necrosis in patients with decreased protein C activity; thus, DOACs that do not suppress protein C production were used in the current case. We have included the corresponding information in-text. (Lines 194-196) PMID: 19615543

  1. Why was the dosage of rivaroxaban 30 mg and not 20 mg as indicated in the drug package insert?

We thank the Reviewer for their question. We reconfirmed that the dosage of rivaroxaban for pulmonary thromboembolism and deep vein thrombosis is 30 mg/day for 3 weeks, according to the drug insert, and then the dose is reduced to 15 mg/day.

  1. Was percutaneous embolectomy considered considering extensive thrombus burden?

We thank the Reviewer for their constructive question. Percutaneous thrombectomy was considered in case sole anticoagulation therapy did not succeed in resolving the venous thrombus; however, treatment with anticoagulation therapy was eventually successful and percutaneous thrombectomy was not necessitated. We have added the corresponding information in-text.(Lines 137-139)

  1. Please check for minor grammatical errors and ensure the tense (present/past) is uniform throughout

We thank the Reviewer for their suggestion. Please rest assured that we have carefully gone through our manuscript again, and we have requested an additional professional English proofreading to make sure it is devoid of any grammatical and language errors.

  1. The arrows in the figures seem to be out of order - please ensure they are reflected appropriately

We thank the Reviewer for pointing this out. We have made the appropriate revisions and ensured the proper positioning of the arrows.
